# Goal-Auxiliary Actor-Critic for 6D Robotic Grasping with Point Clouds

## Abstract

6D robotic grasping beyond top-down bin-picking scenarios is a challenging task. Previous solutions based on 6D grasp synthesis with robot motion planning usually operate in an open-loop setting without considering perception feedback and dynamics and contacts of objects, which makes them sensitive to grasp synthesis errors. In this work, we propose a novel method for learning closed-loop control policies for 6D robotic grasping using point clouds from an egocentric camera. We combine imitation learning and reinforcement learning in order to grasp unseen objects and handle the continuous 6D action space, where expert demonstrations are obtained from a joint motion and grasp planner. We introduce a goal-auxiliary actor-critic algorithm, which uses grasping goal prediction as an auxiliary task to facilitate policy learning. The supervision on grasping goals can be obtained from the expert planner for known objects or from hindsight goals for unknown objects. Overall, our learned closed-loop policy achieves over $90\%$ success rates on grasping various ShapeNet objects and YCB objects in simulation. The policy also transfers well to the real world with only one failure among grasping of ten different unseen objects in the presence of perception noises[1].

## 1 Introduction

Robotic grasping of arbitrary objects is a challenging task. A robot needs to deal with objects it has never seen before, and generates a motion trajectory to grasp an object. Due to the complexity of the problem, majority works in the literature focus on bin-picking tasks, where top-down grasping is sufficient to pick up an object. Both grasp detection approaches (Redmon & Angelova, 2015; Pinto & Gupta, 2016; Mahler et al., 2017) and reinforcement learning-based methods (Kalashnikov et al., 2018; Quillen et al., 2018) are introduced to tackle the top-down grasping problem. However, it is difficult for these methods to grasp objects in environments where 6D grasping is necessary, i.e., 3D translation and 3D rotation of the robot gripper, such as a cereal box on a tabletop or in a cabinet.

While 6D grasp synthesis has been studied using 3D models of objects (Miller & Allen, 2004; Eppner et al., 2019) and partial observations (ten Pas et al., 2017; Yan et al., 2018; Mousavian et al., 2019), these methods only generate 6D grasp poses of the robot gripper for an object, instead of generating a trajectory of the gripper pose to reach and grasp the object. As a result, a motion planner is needed to plan the grasping motion according to the grasp poses. Usually, the planned trajectory is executed in an open-loop fashion since re-planning is expensive, and perception feedback during grasping as well as dynamics and contacts of the object are often ignored, which makes the grasping sensitive to grasp synthesis errors.

In this work, to overcome the limitations in the paradigm of 6D grasp synthesis followed by robot motion planning, we introduce a novel method for learning closed-loop 6D grasping polices from partially-observed point clouds of objects. Our policy directly outputs the control action of the robot gripper, which is the relative 6D pose transformation of the gripper. For the state representation, we adopt an egocentric view with a wrist camera mounted on the robot gripper, which avoids self-occlusion of the robot arm during grasping compared to using an external static camera. Additionally, we aggregate point clouds of the object from previous time steps to avoid ambiguities in the current view and encode the history observations. Our point cloud representation provides richer 3D information for 6D grasping and generalizes better to different objects compared to RGB images.

---

[1]Videos and code are available at `https://sites.google.com/view/gaddpg`

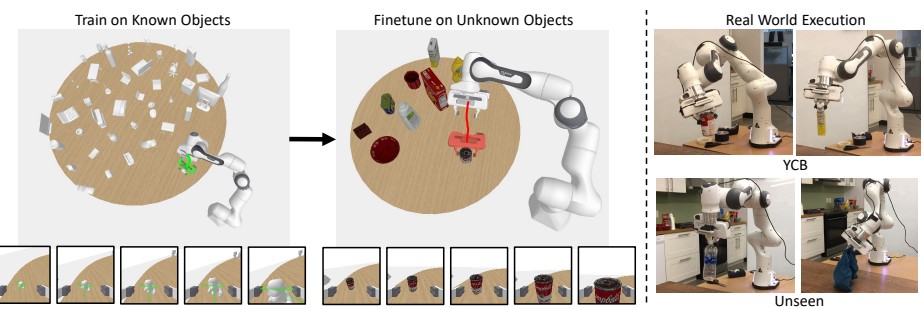

Figure 1: Our method learns the 6D grasping policy with a goal-auxiliary task using an egocentric camera (goals denoted as green forks). We combine imitation learning with a planner (green trajectory) and reinforcement learning for known objects. When finetuning the policy with unknown objects, we use hindsight goals from successful episodes (red trajectory) as supervision. The policy learned in simulation can be successfully applied to the real world for grasping unseen objects.

We propose to combine Imitation Learning (IL) and Reinforcement Learning (RL) in learning the 6D grasping policy. Since RL requires exploration of the state space, the chance of lifting an object is very rare. Moreover, the target object can easily fall down with a bad contact and the ego-view camera may lose the object during grasping. Different from previous works (Song et al., 2020; Young et al., 2020) that specifically design a robot gripper and collect human demonstrations using it, we obtain demonstrations using a joint motion and grasp planner (Wang et al., 2020) in simulation. Consequently, we can efficiently obtain a large number of 6D grasping trajectories using ShapeNet objects (Chang et al., 2015) with the planner. Then we learn the grasping policy based on the Deep Deterministic Policy Gradient (DDPG) algorithm (Lillicrap et al., 2015), which is an actor-critic algorithm in RL that can utilize off-policy data from demonstrations. More importantly, we introduce a goal prediction auxiliary task to improve the policy learning, where the actor and the critic in DDPG are trained to predict the final 6D grasping pose of the robot gripper as well. The supervision on goal prediction comes from the expert planner for objects with known 3D shape and pose. For unknown objects without 3D models available, we can still obtain the grasping goals from successful grasping rollouts of the policy, i.e., hindsight goals. This property enables our learned policy to be finetuned on unknown objects, which is critical for continual learning in the real world. Figure 1 illustrates our setting for learning the 6D grasp polices.

We conduct thorough analyses and evaluations of our method for 6D grasping. We show that our learned policy can successfully grasp unseen ShapeNet objects and unseen YCB objects (Calli et al., 2015) in simulation, and finetuning the policy with hindsight goals for unknown YCB objects improves the grasping success rate. In the real world, we utilized a recent unseen object instance segmentation method (Xiang et al., 2020) to segment the point cloud of an target object, and then applied GA-DDPG for grasping. The learned policy is able to successfully grasp YCB objects in the real world. It only failed one among ten grasping of different unseen objects with perception noises.

Overall, our contributions are three-folds: 1) We propose to use point cloud as our state representation and use demonstrations from a joint motion and grasp planner to learn closed-loop 6D grasping policies. 2) We introduce the Goal-Auxiliary DDPG (GA-DDPG) algorithm for joint imitation and reinforcement learning using goals. 3) We propose to use hindsight goals for finetuning a pre-trained policy on unknown objects without goal supervision.

## 2    RELATED WORK

**Combining Imitation Learning and Reinforcement Learning.** For high-dimensional continuous state-action space with sparse rewards and complex dynamics as in most real-world robotic settings, model-free RL provides a data-driven approach to solve the task (Kalashnikov et al., 2018; Quillen et al., 2018), but it requires a large number of interactions even with full-state information. Therefore, many works have proposed to combine imitation learning (Osa et al., 2018) in RL. For example, Rajeswaran et al. (2017) augment policy gradient update with demonstration data to circumvent reward shaping and the exploration challenge. Zhu et al. (2018) use inverse RL to learn dexterous manipulation tasks with a few human demonstrations in simulation. The closest related works to ours are Vecerik et al. (2017); Nair et al. (2018a) that utilize demonstration data with off-policy RL. Despite the focus on different tasks, the main difference is that our demonstrations are

from an expert planner instead of human demonstrators. Therefore, we can obtain a large number of demonstrations and query the expert planner during training to provide supervision on the on-policy data. Moreover, compared with human demonstrators, expert planners are more likely to generate trajectories that are suitable for robots to execute and have consistency across the state space.

**Goals and Auxiliary Tasks in Policy Learning.** Goals are often used as extra signals to guide policy learning (Kaelbling, 1993). For goal-conditioned policies, Andrychowicz et al. (2017); Ding et al. (2019); Ghosh et al. (2019) make the observation that every trajectory is a successful demonstration of the goal state that it reaches, thereby these methods re-label goals in rollout trajectories for effective learning. However, goals need to be provided in testing, which can be an optimistic assumption for complex tasks. Although many works have been proposed to generate goals or subgoals with deep neural networks (Nair et al., 2018b; Florensa et al., 2018; Sharma et al., 2019), in 6D grasping, estimating the grapsing goals for unseen objects is a challenging task. On the other hand, auxiliary tasks have been employed to improve RL as well. For instance, state reconstruction tasks (Finn et al., 2016), auxiliary control and reward tasks (Jaderberg et al., 2016), pose estimation loss (Zhang et al., 2018) and contrastive loss (Srinivas et al., 2020) have been used to improve representation learning. In our work, we utilize the grasping goal prediction as a natural auxiliary task that requires the policy to predict how to grasp the target object.

**Vision-based Robotic Grasping.** Grasp synthesis can be used in a planning and control pipeline for robotic grasping (Pinto & Gupta, 2016; Mahler et al., 2017; Morrison et al., 2018; Murali et al., 2020). Alternatively, end-to-end policy learning methods (Levine et al., 2018; Kalashnikov et al., 2018; Iqbal et al., 2020) make use of large-scale data to learn closed-loop vision-based grasping. Although RGB images are widely used as the state representation, it requires the policy to infer 3D information from 2D images. Recently, depth and segmentation masks (Mahler et al., 2017), shape completion (Yan et al., 2018), deep visual features (Florence et al., 2019), keypoints (Manuelli et al., 2019), point clouds (Yan et al., 2019), multiple cameras (Akinola et al., 2020), and egocentric views (Song et al., 2020; Young et al., 2020) have been considered to improve the state representation. Motivated by these methods, we utilize the aggregated point clouds of an object from an egocentric camera as our state representation. Moreover, we explicitly learn a policy for the large action space in 6D grasping instead of using value-based representations as in most top-down grasping methods.

## 3 BACKGROUND

**Markov Decision Process.** A Markov Decision Process (MDP) is defined using the tuple: $\mathcal{M} = \{\mathcal{S}, \mathcal{R}, \mathcal{A}, \mathcal{O}, \mathcal{P}, \rho_0, \gamma\}$. $\mathcal{S}$, $\mathcal{A}$, and $\mathcal{O}$ represent the state, action, and observation space. $\mathcal{R} : \mathcal{S} \times \mathcal{A} \to \mathbb{R}$ is the reward function. $\mathcal{P} : \mathcal{S} \times \mathcal{A} \to \mathcal{S}$ is the transition dynamics. $\rho_0$ is the probability distribution over initial states and $\gamma = [0, 1)$ is a discount factor. Let $\pi : \mathcal{S} \to \mathcal{A}$ be a policy which maps states to actions. In the partially observable case, at each time $t$, the policy maps a partial observation $o_t$ of the environment to an action $a_t = \pi(o_t)$. Our goal is to learn a policy that maximizes the expected cumulative rewards $\mathbb{E}_\pi[\sum_{t=0}^{\infty} \gamma^t r_t]$, where $r_t$ is the reward at time $t$. The Q-function of the policy for a state-action pair is $Q(s, a) = \mathcal{R}(s, a) + \gamma \mathbb{E}_{s', \pi}[\sum_{t=0}^{\infty} \gamma^t r_t | s_0 = s']$, where $s'$ represents the next state of taking action $a$ at state $s$ according to the transition dynamics.

**Deep Deterministic Policy Gradient.** Deep Deterministic Policy Gradient (DDPG) (Lillicrap et al., 2015) is an actor-critic algorithm that uses off-policy data and temporal difference learning, and has successful applications in continuous control (Popov et al., 2017; Vecerik et al., 2017). Specifically, the actor in DDPG learns the policy $\pi_\theta(s)$, while the critic approximates the Q-function $Q_\phi(s, a)$, where $\theta$ and $\phi$ denote the parameters of the actor and the critic, respectively. A replay buffer of transitions $\mathcal{D} = \{(s, a, r, s')\}$ is maintained during training, and examples sampled from it are used to optimize the actor-critic. DDPG minimizes the following Bellman error with respect to $\phi$ :

$$L(\phi) = \mathbb{E}_{(s,a,r,s') \sim \mathcal{D}} \Big[ \frac{1}{2} \Big( Q_\phi(s, a) - \big( r + \gamma Q_\phi(s', \pi_\theta(s')) \big) \Big)^2 \Big]. \tag{1}$$

Then the deterministic policy $\pi_\theta$ is trained to maximize the learned Q-function with objective $\max_\theta \mathbb{E}_{s \sim \mathcal{D}}(Q_\phi(s, \pi_\theta(s)))$, which resembles a policy evaluation and improvement schedule.

## 4 LEARNING 6D GRASPING POLICIES

We consider the task of grasping an arbitrary object in a closed-loop manner, where partial observations come from a wrist camera mounted on the end-effector of a robot. Our goal is to learn a 6D

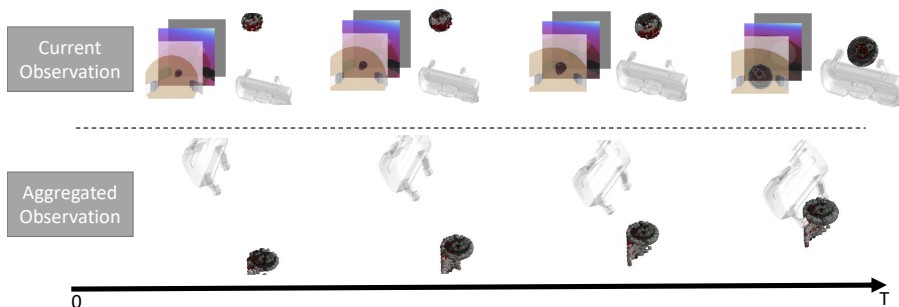

Figure 2: Illustration of our point aggregation procedure. We show the RGB, depth, foreground mask and point cloud at time $t$, and the aggregated point cloud up to time $t$.

grasping policy $\pi : s_t \rightarrow a_t$ that maps the state $s_t$ at time $t$ to an action $a_t$. To simplify the setting, we still use $s$ to represent the input of the policy even though observations are used. At time $t$, the action $a_t$ is parametrized by the relative 3D translation and the 3D rotation of the robot end-effector. Therefore, our learned visuomotor policy represents an operational space controller for 6D grasping.

## 4.1 6D GRASPING FROM POINT CLOUDS

We utilize 3D point clouds of objects to represent the states, which can be computed using depth images and foreground masks of the objects. To better utilize the 3D information and resolve ambiguities of the current view due to local geometry, we propose to aggregate point clouds as the gripper moves in time for our state representation. Figure 2 illustrates our point aggregation procedure.

In order to aggregate 3D points from different time steps, we need to transform them into a common coordinate system. We choose the coordinate of the robot base to aggregate the 3D points. Specifically, at time $t$, let $X_t^C \in \mathbb{R}^{n_t \times 3}$ denote the 3D point cloud of the target object from the wrist camera, where $n_t$ represents the number of points at time $t$. Since the camera is attached to the robot hand, we first transform $X_t^C$ to the end-effector frame using the camera extrinsics denoted as $X_t$. Using forward kinematics, we can compute the transformation of the end-effector in the robot base as $\mathcal{T}_t \in \mathbb{SE}(3)$. Then we can transform the point cloud to the robot base by $\mathcal{T}_t^{-1}(X_t)$. Finally, our state representation at time $t$ is $s_t = \mathcal{T}_t\left(\cup_{i=0}^{t} \mathcal{T}_i^{-1}(X_i)\right)$, i.e., aggregating the point clouds in the robot base up to time $t$ and then transforming the points to the end-effector frame at time $t$. Using 3D points suffers less from the sim-to-real gap.

## 4.2 DEMONSTRATIONS FROM A JOINT MOTION AND GRASP PLANNER

To obtain demonstrations for 6D grasping, we utilize the recent Optimization-based Motion and Grasp Planner (OMG Planner) (Wang et al., 2020) as our expert. Given a planning scene and a set of pre-defined grasps on a target object from a grasp planner (Miller & Allen, 2004; Eppner et al., 2019), the OMG Planner plans a trajectory of the robot to grasp the object. Depending on the initial configuration of the robot, the OMG Planner selects different grasps as the goal. This property enables us to learn a policy that grasps objects in different ways according to the initial pose of the robot gripper, which helps alleviate the multi-modality problem in 6D grasping.

Let $\xi = (\mathcal{T}_0, \mathcal{T}_1, \ldots, \mathcal{T}_T)$ be a trajectory of the robot gripper pose generated from the OMG planner to grasp an object, where $\mathcal{T}_t \in \mathbb{SE}(3)$ is the gripper pose in the robot base frame at time $t$. Then the expert action at time $t$ can be computed as $a_t^* = \mathcal{T}_{t+1}\mathcal{T}_t^{-1}$, which is the relative transformation of the gripper between $t$ and $t + 1$. Furthermore, $\mathcal{T}_T$ from the OMG Planner is a successful grasp pose of the object. We define the expert goal at time $t$ as $g_t^* = \mathcal{T}_T\mathcal{T}_t^{-1}$, which is the relative transformation between the current gripper pose and the final grasp pose. Finally, the state $s_t$ can be obtained by computing the aggregated point cloud as in Section 4.1. In this way, we construct an offline dataset of demonstrations from the OMG Planner as $\mathcal{D}_{\text{offline}} = \{(s_t, a_t^*, g_t^*, s_t')\}$, where $s_t'$ is the next state.

## 4.3 BEHAVIOR CLONING AND DAGGER

Using the demonstration dataset $\mathcal{D}_{\text{offline}}$, we can already train a policy network $\pi_\theta$ by applying Behavior Cloning (BC) to predict an action from a state. Different loss functions on $\mathbb{SE}(3)$ can be used.

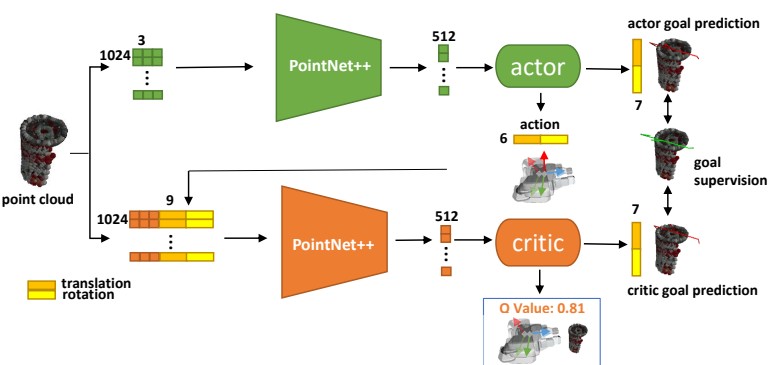

Figure 3: Our network architecture with the PointNet++ network (Qi et al., 2017) for feature extraction. Both the actor and the critic are regularized to predict grasping goals as auxiliary tasks. Note that we use quaternions in grasping goal prediction and Euler angles for the action.

In particular, we adopt the point matching loss function to jointly optimize translation and rotation:

$$L_{\text{POSE}}(\mathcal{T}_1, \mathcal{T}_2) = \frac{1}{|X_g|} \sum_{x \in X_g} \|\mathcal{T}_1(x) - \mathcal{T}_2(x)\|_1, \tag{2}$$

where $\mathcal{T}_1, \mathcal{T}_2 \in \mathbb{SE}(3)$, $X_g$ denotes a set of pre-defined 3D points on the robot gripper, and L1 norm is used. Let $a^\theta = \pi_\theta(s)$ be an action predicted from the policy network and $a^* = \pi^*(s)$ be an action from the expert, the behavior cloning loss is $L_{\text{BC}}(a^\theta, a^*) = L_{\text{POSE}}(a^\theta, a^*)$.

We apply DAGGER (Ross et al., 2011) to augment supervisions for rollouts from the learner. Given a sampled initial state $s_0$, the current policy $\pi_\theta$ can roll out a trajectory of states $s_0, ..., s_T$ and actions $a_0, ..., a_T$ with end-effector poses $\hat{\mathcal{T}}_0, ..., \hat{\mathcal{T}}_T$. The expert then treats each state as the initial state and generates a trajectory to grasp the object $\xi(s_t) = (\hat{\mathcal{T}}_t, \mathcal{T}_{t+1}, ..., \mathcal{T}_T)$. We use the first action from the expert as supervision $a_t^* = \pi^*(s_t) = \mathcal{T}_{t+1}\hat{\mathcal{T}}_t^{-1}$ to correct deviation of the learner from $\xi(s_t)$. A corresponding expert goal $g_t^* = \mathcal{T}_T\hat{\mathcal{T}}_t^{-1}$ can also be obtained for state $s_t$. These action and goal labels are used to construct a dataset $D_{\text{dagger}} = \{(s_t, a_t, a_t^*, g_t^*, s_t')\}$.

### 4.4 GOAL-AUXILIARY DDPG

To handle contact-rich scenarios that are rare in the expert trajectories, we further improve the policy learning by combining RL with expert demonstrations. In our RL setting, each episode ends when the agent attempts a grasp or reaches a maximum horizon $T$. The sparse reward $r_t$ is an indicator function given at the end of the episode denoting if the target object is lifted or not.

Similar to DAGGER, we collect on-policy data for DDPG training. The only difference is that we do not have expert demonstration for the action $a_t^*$ and goal $g_t^*$ for a state $s_t$ from the learner policy. Instead, we first find the nearest goal from a goal set $\mathcal{G}$ by $\tilde{g}_t = \arg\min_{g \in \mathcal{G}} \|g - \mathcal{T}_t\|$, where $\mathcal{T}_t$ is the pose of the robot gripper at time $t$. Then the heuristics goal for $s_t$ is $g_t = \tilde{g}_t \mathcal{T}_t^{-1}$. Note that the goal set $\mathcal{G}$ is only available if we have the 3D shape and pose of the target object. Finally, we can construct an on-policy dataset for DDPG $\mathcal{D}_{\text{ddpg}} = \{(s_t, a_t, g_t, r_t, s_t')\}$, where $r_t$ is the reward at state $s_t$ and $s_t'$ is the next state of $s_t$ from the on-policy rollout. The replay buffer of DDPG training is $\mathcal{D} = \mathcal{D}_{\text{offline}} \cup \mathcal{D}_{\text{dagger}} \cup \mathcal{D}_{\text{ddpg}}$, where we augment $\mathcal{D}_{\text{offline}}$ and $\mathcal{D}_{\text{dagger}}$ with the sparse reward.

Our network architecture for training the actor-critic is shown in Figure 3. We introduce two auxiliary tasks for the actor-critic to predict goals, which help to learn the policy and the Q-function by adding goal supervision. Given a data sample $(s, a, g, r, s')$ from the replay buffer, the critic loss is

$$L_{Q_\phi}(s, a, r, s', g, g^\phi) = \frac{1}{2}\Big[Q_\phi(s, a) - \Big(r + \gamma Q_{\phi'}\big(s', \pi_{\theta'}(s') + \epsilon\big)\Big)\Big]^2 + L_{\text{AUX}}(g, g^\phi), \tag{3}$$

where $g^\phi$ is the predicted goal, $Q_{\phi'}$ and $\pi_{\theta'}$ are the target networks and $\epsilon$ is a pre-defined clipped noise as in TD3 (Fujimoto et al., 2018). The auxiliary loss is $L_{\text{AUX}}(g, g^\phi) = L_{\text{POSE}}(g, g^\phi)$.

Given a data sample $(s, a^*, g)$ from the replay buffer, the loss function for the actor is defined as

$$L_{\pi_\theta}(s, a^*, a^\theta, g, g^\theta) = \lambda L_{BC}(a^*, a^\theta) + (1-\lambda)L_{\text{DDPG}}(s, a^\theta) + L_{\text{AUX}}(g, g^\theta), \tag{4}$$

where $a^\theta$ and $g^\theta$ are the action and the goal predicted from the actor, respectively, and $\lambda$ is a weighting hyper-parameter to balance the losses from the expert and from the learned value function. The BC loss $L_{BC}(a^*, a^\theta) = L_{\text{POSE}}(a^*, a^\theta)$ as defined before, which prevents the learner to move too far away from the expert policy. The deterministic policy loss is defined as $L_{\text{DDPG}}(s, a^\theta) = -Q_\phi(s, a^\theta)$, and the auxiliary loss is $L_{\text{AUX}}(g, g^\theta) = L_{\text{POSE}}(g, g^\theta)$. Note that the expert action $a^*$ is only available if the data sample is from $\mathcal{D}_{\text{offline}}$ or $\mathcal{D}_{\text{dagger}}$. Otherwise, we do not include the BC loss in Eq. (4). Even though our method can be used with any actor-critic algorithms, in practice, we choose TD3 for its improved performance over vanilla DDPG.

## 4.5 HINDSIGHT GOALS FOR FINETUNING ON UNKNOWN OBJECTS

If we would like to deploy the trained policy to the real world, the agent needs to potentially deal with unknown objects and different dynamics. In this case, we cannot obtain neither expert demonstrations nor grasping goals from a pre-defined goal set. We introduce using hindsight goals to finetune our policy on unknown objects for self-supervised continual learning.

Based on the current policy $\pi_\theta$, each learner episode rolls out a trajectory of state-actions with end-effector poses $\hat{\mathcal{T}}_0, ..., \hat{\mathcal{T}}_T$. If the grasp in the episode is successful, we know that $\hat{\mathcal{T}}_T$ is a success grasping pose for the target object, which is also known as the hindsight goal. Then we can compute the grasp goal for state $s_t$ in the episode as $\hat{g}_t = \hat{\mathcal{T}}_T \hat{\mathcal{T}}_t^{-1}$, which is used to supervise the actor and the critic via the goal auxiliary tasks. By using hindsight goals, we construct a dataset using on-policy rollouts on unknown objects $\mathcal{D}_{\text{hindsight}} = \{(s_t, a_t, \hat{g}_t, r_t, s_t')\}$, where $\hat{g}_t$ is the hindsight goal. The new loss functions for the critic and the actor can be rewritten as

$$L_{Q_\phi}(s, a, r, s', \hat{g}, g^\phi, r_T) = \frac{1}{2}\left[Q_\phi(s, a) - \left(r + \gamma Q_{\phi'}\left(s', \pi_{\theta'}(s') + \epsilon\right)\right)\right]^2 + L_{\text{AUX}}(\hat{g}, g^\phi) \cdot r_T$$

$$L_{\pi_\theta}(s, a^*, a^\theta, \hat{g}, g^\theta, r_T) = \lambda L_{BC}(a^*, a^\theta) - (1 - \lambda)L_{\text{DDPG}}(s, a^\theta) + L_{\text{AUX}}(\hat{g}, g^\phi) \cdot r_T, \quad (5)$$

where $r_T \in \{0, 1\}$ is the binary reward for the last state in the episode. Therefore, we only have the goal auxiliary losses for successful episodes during finetuning on unknown objects. We finetune the pretrained actor-critic network on the dataset $\mathcal{D} = \mathcal{D}_{\text{offline}} \cup \mathcal{D}_{\text{hindsight}}$. Note that we cannot collect extra data for $\mathcal{D}_{\text{dagger}}$ and $\mathcal{D}_{\text{ddpg}}$ with unknown objects, but we can potentially use the ones that are already collected. Our algorithm of using hindsight goals is described in Appendix Section A.

## 5 EXPERIMENTS

We conduct experiments and present our findings to the following questions. (1) For BC with supervised learning, can using point clouds achieve better generalization for grasping unseen objects than using images? (2) With RL, how can different strategies of using grasping goals affect the performance? (3) Can our method further improve on unknown objects using hindsight goals, even if the dynamics has changed? We also conduct ablation studies on different components in our method, a comparison with an open-loop baseline and real-world experiments for 6D grasping.

**Task Environment.** We experiment with the Franka Emika Panda arm, a 7-DOF arm with a parallel gripper. We use ShapeNet (Chang et al., 2015) and YCB objects (Calli et al., 2015) as our object repository. A task scene is generated by dropping a sampled target object with a random pose on a table in the PyBullet Simulator (Coumans & Bai, 2016–2019). The maximum horizon for the policy is $T = 30$. An episode is terminated once a grasp is completed. The observation image size is $112 \times 112$ in PyBullet. More environment details can be found in Appendix Section B.1.

**Training and Testing.** We use approximately $1,500$ ShapeNet objects from 169 different classes for training, where each object has 100 pre-computed grasps from Eppner et al. (2019) for planning. For testing, we use 9 selected YCB objects with 10 scenes per object and 138 holdout ShapeNet objects within 157 scenes. We train each model three times with different random seeds. We run each YCB scene 5 times and each ShapeNet scene 3 times and then compute the mean grasping success rate. More details on network and train-test can be found in Appendix Section B.2 and B.3.

## 5.1 BEHAVIOR CLONING WITH EXPERT SUPERVISION

We first conduct experiments with BC to investigate the effect of using different state representations. Table 1 presents the grasping success rates. When using images, there are three variations:

| Input | Method | | | | Test | |
|---|---|---|---|---|---|---|
| | Offline | Online | DAGGER | Goal-Auxiliary | ShapeNet | YCB |
| RGB | ✓ | | | | 0.436 | 0.452 |
| RGB | | ✓ | | | 0.506 | 0.515 |
| RGB | | ✓ | ✓ | | 0.521 | 0.523 |
| RGB | | ✓ | ✓ | ✓ | 0.543 | 0.581 |
| RGBD | ✓ | | | | 0.453 | 0.463 |
| RGBD | | ✓ | | | 0.587 | 0.653 |
| RGBD | | ✓ | ✓ | | 0.604 | 0.671 |
| RGBD | | ✓ | ✓ | ✓ | 0.680 | 0.717 |
| RGBDM | ✓ | | | | 0.483 | 0.504 |
| RGBDM | ✓ | | | ✓ | 0.558 | 0.522 |
| RGBDM | | ✓ | | | 0.674 | 0.667 |
| RGBDM | | ✓ | ✓ | | 0.715 | 0.723 |
| RGBDM | | ✓ | ✓ | ✓ | 0.752 | 0.726 |
| Point | ✓ | | | | 0.736 | 0.656 |
| Point | ✓ | | | ✓ | 0.733 | 0.673 |
| Point | | ✓ | | | 0.727 | 0.721 |
| Point | | ✓ | | ✓ | 0.723 | 0.727 |
| Point | | ✓ | ✓ | | 0.758 | 0.772 |
| Point | | ✓ | ✓ | ✓ | **0.796** | **0.785** |

Table 1: Success rates of different models trained by behavior cloning with expert supervision.

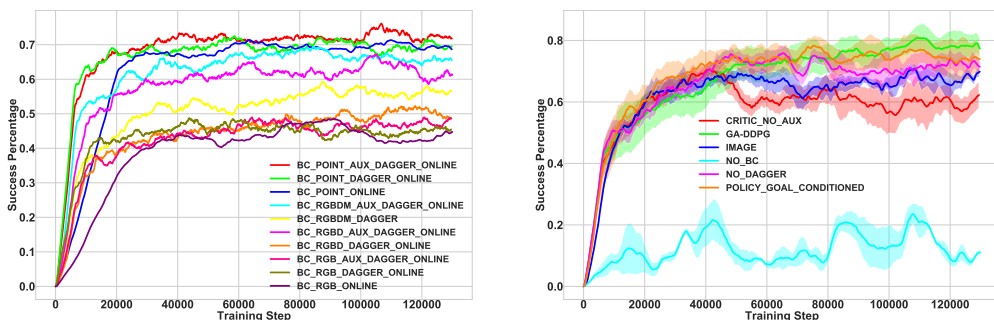

Figure 4: (a) Learning curves of different BC models. (b) Learning curves of different RL models. The mean and the 1 standard deviation range are plotted for each model.

"RGB", "RGBD" and "RGBDM" indicating whether depth (D) or foreground mask (M) of the object are used or not. Depth and/or mask are concatenated with the RGB image, and ResNet (He et al., 2016) is used for feature extraction. "Point" in Table 1 indicates the aggregated point cloud representation we propose. "Offline" means a fixed-size offline dataset $\mathcal{D}_{\text{offline}}$ is used for training which contains 50,000 data points from expert demonstrations. "Online" means the expert planner is running in parallel with training, which keeps adding new data to the dataset. "DAGGER" uses on-policy rollout data from the learner policy with expert supervision $\mathcal{D}_{\text{dagger}}$, and "Goal-Auxiliary" indicates whether we add the grasping goal prediction task or not in BC training.

From Table 1, we can see that: i) using the aggregated point clouds achieves better performance compared to using images of the current view. This indicates that the 3D features in object point clouds generalizes better for 6D grasping. ii) Adding depth or foreground mask to the image-based representation improves the performance. iii) "Online" is overall better than "Offline" by utilizing more data for training. iv) Both DAGGER and adding the auxiliary task improve grasping success rate. Figure 4(a) shows the learning curves of several BC models.

## 5.2 GOAL-AUGMENTED REINFORCEMENT LEARNING

We evaluate our goal-auxiliary DDPG algorithm for 6D grasping, where the dataset $\mathcal{D} = \mathcal{D}_{\text{offline}} \cup \mathcal{D}_{\text{dagger}} \cup \mathcal{D}_{\text{ddpg}}$ is constructed for training. We also consider another common strategy of using goals for comparison: goal-conditioned policies such as in Andrychowicz et al. (2017); Nair et al. (2018a); Ding et al. (2019), where a goal is concatenated with the state as the input for the network.

| Policy / Critic | None | | Goal-Auxiliary | | Goal-Conditioned | | Both | |
|---|---|---|---|---|---|---|---|---|
| | ShapeNet | YCB | ShapeNet | YCB | ShapeNet | YCB | ShapeNet | YCB |
| None | 0.806 | 0.710 | 0.773 | 0.728 | 0.686 | 0.607 | 0.767 | 0.766 |
| Goal-Auxiliary | 0.846 | 0.828 | **0.913** | **0.882** | 0.834 | 0.813 | 0.878 | 0.841 |
| Goal-Conditioned | 0.816 | 0.737 | 0.793 | 0.761 | 0.703 | 0.632 | 0.821 | 0.779 |
| Both | 0.868 | 0.825 | 0.879 | 0.831 | 0.845 | 0.817 | 0.869 | 0.823 |

Table 2: Evaluation on success rates of two strategies of using goals in RL with DDPG.

| Test | No BC | Offline | No DAGGER | Image | No Aggr. | Late Fusion | No Expert Init. | L2 Loss | Final |
|------|-------|---------|-----------|-------|----------|-------------|-----------------|---------|-------|
| ShapeNet | 0.027 | 0.723 | 0.839 | 0.837 | 0.837 | 0.791 | 0.877 | 0.818 | **0.913** |
| YCB | 0.038 | 0.637 | 0.853 | 0.789 | 0.835 | 0.821 | 0.866 | 0.792 | **0.882** |

Table 3: Ablation studies on different components in our method with grasping success rates.

In this case, in order to use goals in testing, we need to train a separate network to predict goals from states. Table 2 displays the evaluation results, where we test different combinations of adding the goal-auxiliary task or using the goal-conditioned input to the actor and the critic.

From Table 2, we can see that: i) using the goal-auxiliary loss for the critic significantly improves the performance since it regularizes the Q-learning. ii) The goal-conditioned policies perform worse than the goal-auxiliary counterparts. Mainly because predicting the 6D grasping goals itself is a challenging problem and treating grasping as reaching the target goal is simplistic. When the predicted goal is not accurate, it affects the grasping success. We also observe some instability in goal-conditioned policy training. iii) Adding the goal-auxiliary task to both the actor and the critic achieves the best performance. iv) Comparing to the supervised BC training, our GA-DDPG algorithm further improves the policy, especially in those contact-rich scenarios. Imitation learning alone cannot handle these contacts with the target object before closing the fingers since these are rarely seen in the expert demonstrations.

### 5.3 FINETUNING WITH HINDSIGHT GOALS

Our policy trained on ShapetNet objects achieves 91.3% and 88.2% success rate for grasping unseen ShapeNet and unseen YCB objects, respectively, as shown in Table 2. We can further improve the policy using RL with unseen objects. To resemble real-world RL, we assume that the 3D shape and pose of the YCB objects are not available. Therefore, there is no expert supervision on these objects. We finetune the policy using the dataset $\mathcal{D} = \mathcal{D}_{\text{offline}} \cup \mathcal{D}_{\text{hindsight}}$ with hindsight goals. We see that the success rate is improved from 88.2% to 93.5% after finetuning. We also experiment with changing the dynamics in simulation by decreasing the lateral frictions of the object, which makes object easier to slide during contacts. In this case, the success rate of the pre-trained policy reduces to 81.6%. After finetuning with hindsight goals, the policy achieves 88.5% success rate, which demonstrates the effectiveness of using hindsight goals for finetuning.

### 5.4 ABLATION STUDIES

We conduct ablation studies on several design choices in GA-DDPG as shown in Table 3. We can see that: i) RL without BC failed to learn a useful policy due to the high-dimensional state-action space and sparse reward. ii) Both online interaction for adding data and DAGGER help. iii) Our aggregated point cloud representation is better than using "RGBDM" images or single-frame point clouds. iv) Early fusion of the state-action in the critic is better than late fusion, i.e., concatenating features of state and action. v) We also use expert-aided exploration by rolling out expert plans for a few steps to initiate the start states during learner rollouts and denote it as "Expert Init.", which is beneficial. vi) The point matching loss defined in Eq. (2) is better than a L2 loss on translation and rotation. Overall, we observe that these design choices both stabilize training and improve the performance. Figure 4(b) shows the learning curves of several RL models. Ablation experiments on grasp prediction performance can be found in Appendix Section C.

### 5.5 COMPARISON WITH AN OPEN-LOOP BASELINE

We compare our learned closed-loop policy with a traditional manipulation pipeline consisting of grasp detection from partial point clouds, motion planning to generate trajectories to reach grasps, and open-loop execution. Specifically, we use a state-of-the-art grasp detection method, 6-DOF GraspNet (Mousavian et al., 2019), to generate a set of grasps. We then use the top 100 feasible grasps according to their grasp quality scores as the grasp set for the OMG planner. The planner selects a grasp and generates a trajectory to reach the grasp. Finally, the generated trajectory is executed in PyBullet for grasping. This open-loop method achieves 75.7% success rate in the simulator on grasping YCB objects. Our learned closed-loop policy from GA-DDPG achieves 91.3% success rate without finetuning on YCB objects, which is significantly better than this open-loop baseline. Our finetuned policy matches the performance of the open-loop expert with perfect grasps at around 94%. This experiment demonstrates that closed-loop 6D grasping is critical as open-loop methods are heavily affected by the grasp detection performance.

| Object | Cracker | Sugar | Tomato Soup | Mustard | Meat Can | Bleach | Bowl | Mug | Foam Brick | Unseen | Mean |
|---|---|---|---|---|---|---|---|---|---|---|---|
| Fix | 5/5 | 5/5 | 3/5 | 4/5 | 3/5 | 4/5 | 4/5 | 4/5 | 5/5 | 9/10 | 0.84 |
| Varying | 4/5 | 5/5 | 2/5 | 4/5 | 2/5 | 4/5 | 5/5 | 4/5 | 4/5 | 9/10 | 0.78 |

Table 4: Real-world grasping results. The numbers of successful grasps among trials are presented.

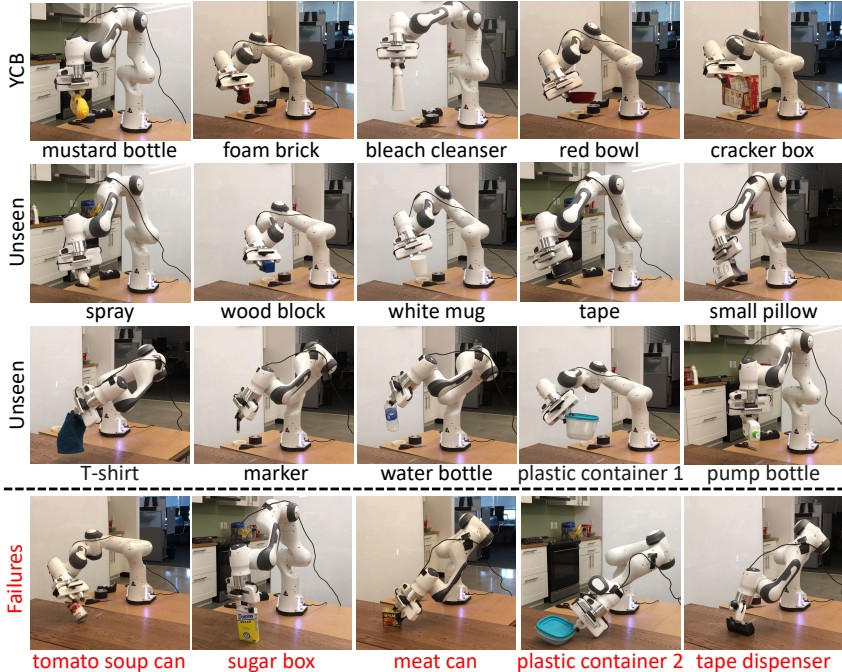

Figure 5: Success and failure of real-world 6D grasping using our policy trained in simulation.

## 5.6 REAL-WORLD EXPERIMENTS

Finally, we conduct real-world grasping experiments with GA-DDPG. We directly applied our policy learned in PyBullet to the real world. To obtain the segmented point cloud of an object, we utilized an unseen object instance segmentation method (Xiang et al., 2020). The policy outputs the relative 6D pose transformation of the robot end-effector at each time step, which is used to control the robot for grasping. For the 9 YCB objects, the robot grasped each object 5 times in a setting with a fixed robot initial pose and a setting with varying robot initial poses. In addition, 10 unseen objects with various shapes and materials are tested. Table 4 presents the numbers of successful grasps among trials. The learned policy achieves around 80% success rate. It only failed for one object among the ten unseen objects in both settings. Figure 5 shows some successful grasping examples with different objects. The last row in Figure 5 shows some examples of failure cases. We see two main categories of failures. The first one is due to inaccurate grasp poses from the policy. The second one is due to slippery and mismatch in physics between simulation and the real world, where the same hand pose can succeed in simulation but fail in the real world. Making the simulation better matching the real world or finetuning the policy in the real world would improve the policy.

## 6 CONCLUSION

We have proposed a novel method for efficiently learning 6D grasping control polices from point clouds. Our method combines demonstrations from an expert motion and grasp planner with RL to explore on-policy data. It learns a closed-loop control policy which takes into account dynamics and contacts with objects in grasping. We introduce the goal-auxiliary DDPG algorithm for policy learning, which uses the goal prediction as an auxiliary task to improve the performance of actor and critic. We demonstrate that our method trained on ShapeNet objects can be successfully applied to grasping unseen objects. Furthermore, we show that our policy can be finetuned on unknown objects using hindsight goals to achieve continual learning. We also demonstrated that our policy trained in simulation can be applied to the real world. It achieves robustness and generalization across diverse objects and initial robot poses in the real world for grasping unseen objects. As for future work, we plan to extend our method to 6D grasping in cluttered scenes.

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

# APPENDIX

## A    HINDSIGHT GOAL AUXILIARY TRAINING ALGORITHM

---

**Algorithm 1:** Hindsight Goal Auxiliary Training (policy $\pi_\theta$, dataset $\mathcal{D}_{\text{hindsight}}$, critic $Q_\phi$)

---
**for** $i = 0, ..., N$ **do**
  Execute $\pi_\theta$ in environment for an episode $\tau$
  Log trajectory $\tau = (s_0, a_0, r_0..., s_T, a_T, r_T)$
  Compute $\hat{g}_t$ from $\tau$ and augment goals to the dataset $\mathcal{D}_{\text{hindsight}}$
  Optimize the $Q_\phi, \pi_\theta$ with Eq. (5)
**end**

---

Compared with the common relabeling strategy for goal-conditioned policies (Andrychowicz et al., 2017; Ding et al., 2019; Ghosh et al., 2019), we do not modify the reward of an episode. While using every hindsight goal might allow the robot to reach diverse goals, it hurts grasping performance if we use unsuccessful grasp poses as goals. Moreover, we do not treat grasping as a goal-reaching task since solving the grasp detection problem is also a challenge. It will also ignore contacts during grasping and require a perfect grasp pose as input at inference time.

## B    IMPLEMENTATION DETAILS

### B.1    TASK ENVIRONMENT DETAILS

The expert planner is tuned to have around $95\%$ success rates on both YCB and ShapeNet objects. For point aggregation at time step $t$, we sample $m_t$ points among the $n_t$ observed points uniformly to limit the number of points. Note that a closer view of the object leads to a larger foreground mask. Therefore, $n_t$ increases as $t$ goes from 0 to $T$. In order to balance points sampled from different time steps, we aggregate observed point clouds in an exponential decaying schedule. Specifically, $m_t = \lceil \alpha^t n_t \rceil$, where $\alpha = 0.95$. The network input contains 1024 sampled points from state $s_t$.

Compared to the YCB dataset which composes of objects that are manipulated in daily life, some objects in the ShapeNet repository such as Shelves, Desks, and Beds do not resemble real-world tabletop objects. Therefore we resize them to be graspable by the Panda gripper and use the same object physics for all the objects (Eppner et al., 2019; Tobin et al., 2018). We randomize initial states $\rho_0$ from an upper-sphere pose distribution facing toward the target object in a distance from $0.2$m to $0.5$m. Compared to the restricted top-down grasping, the diversity of the initial arm configurations and object shapes and poses forces the policy to learn grasping in the 6D space. When RGB images are used, we apply domain randomization during training by randomly changing the textures.

For the task setup, we have also tried using RGB images from an overhead static camera as input and joint positions/velocities as the actions. For 6D grasping, we believe that using the ego motion from a wrist camera generalizes better to unseen objects in different backgrounds and suffers less from occlusions during grasping. Moreover, the aggregated point cloud is suitable for sim-to-real transfer since it has no reliance on color information and camera parameters, and often shows robustness to depth noises from sensors. It also allows motion based on history observations even when the current timestep has limited sensory feedback.

### B.2    TRAINING AND TESTING DETAILS

The dataset $\mathcal{D}_{\text{offline}}$ contains 50,000 data points in successful episodes from expert demonstrations and extra 120,000 failure data points in the case of offline RL in our ablation studies. These data contain a few rollouts for each object to resemble the real-world setting where the demonstration data has a fixed size. Our online setting has around 3 million interaction steps. The replay buffer contains $2 \cdot 10^6$ transitions for points and $2 \cdot 10^5$ transitions for images due to the CPU memory limit.

During each iteration of the DDPG training, we sample 14 parallel rollouts of the actor to collect experiences. We then perform 20 optimization steps on a mini-batch of size 250 which has $70\%$ of

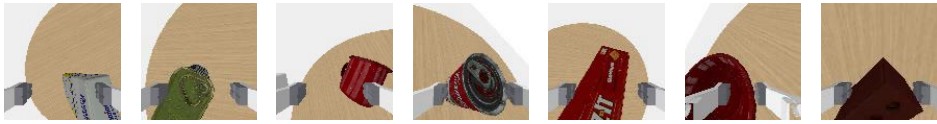

Figure 6: Complex contacts between the robot gripper with different objects during grasping.

the data from the on-policy buffer and the rest from expert buffer. We add decaying uniform noises from 3cm, 8 degrees to 0.5cm, 1 degree for the exploration on translation and rotation, respectively. During expert rollouts, we randomly apply motion perturbations to increase the diverse state space explorations. The balancing ratio of the BC loss and the DDPG loss is $\lambda = 0.2$. We use a discount factor of $\gamma = 0.95$ for the MDP. We adopt the TD3 algorithm (Fujimoto et al., 2018), and the policy network is updated once every two Q-function updates. The target network of the actor and the first target network of the critic are updated every iteration with a Polyak averaging of $0.999$. The second target network used in Q-learning uses $3,000$ update step delay (Kalashnikov et al., 2018). In the pose loss $L_{\text{POSE}}$, the $X_g$ contains 6 keypoints on the finger tips, finger bases, and the palm base of the robot gripper.

### B.3 NETWORK ARCHITECTURE DETAILS

The feature extractor backbone for point clouds is PointNet++ (Qi et al., 2017) with a model size around 10M, while the backbone network for images is ResNet18 (He et al., 2016) which has much more parameters with a model size of 45M. The light-weighted structure of PointNet++ supports efficient computation in distributed RL, large batch size, and less overfitting with faster convergence. We note that other 3D networks should also be applicable. For ResNet18 with depth and foreground masks, we concatenate the depth image and the mask to the RGB image. The weights of the first convolution layer corresponding to the two additional channels are initialized with all zeros.

The remaining timestep $T - t$ is concatenated in the features to provide information about the remaining horizon. The actor network and the critic network are both implemented by a multi-layer perceptron with three fully-connected layers using the ReLU activation function. We use Adam as the optimizer for training and decrease the learning rate in a piece-wise step schedule. We use separate feature extractors for the actor and the critic. For finetuning with hindsight goals, we fix the weights of the feature extractor backbones, and only finetune the actor and critic networks.

### B.4 DESIGN LIMITATIONS

The underlying assumption for using point aggregation and hindsight goals in our method is that the object pose does not change dramatically due to robot interaction, which we find quite applicable for precision grasping. Moreover, we use the depth region between the two fingertips to determine if the robot should close the fingers and retract the arm to complete a grasp. In the real world where depth and mask perception are limited as the end-effector is close to the target, we use the aggregated points inside the cage region of the gripper as well as the time step $t$ to determine termination. Although this heuristic would not allow re-grasp or other learned grasp closure behavior, we find it generalized and robust enough to simplify the task. In the simulator, our value function in the actor-critic can improve the policy, especially in those contact-rich scenarios (Figure 6) even physics parameters are varied. However, when deploying the learned policy to the real world, we still observe the sim-to-real domain gap in contact modeling. Finally, while an action representation of end-effector delta pose makes the policy learning easier by removing some of the complexity of the configuration space and dynamics, it is less suitable for high-frequency control domains.

## C GRASP PREDICTION PERFORMANCE

Since our method involves grasping goal prediction, we evaluate the grasp detection quantitatively and qualitatively. In this experiment, we train a policy network using BC on the offline dataset and use point clouds as our input.

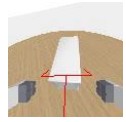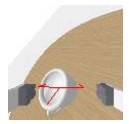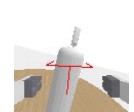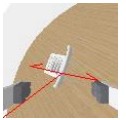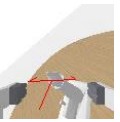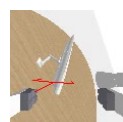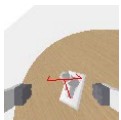

Figure 7: Grasp prediction examples on several ShapeNet objects. The red fork denotes the gripper pose predicted from our policy network.

| Method | Grasp-Translation (m) | | | | Grasp-Rotation (°) | | | | Grasp-Point-Distance (m) | | | |
|---|---|---|---|---|---|---|---|---|---|---|---|---|
| | ShapeNet | | YCB | | ShapeNet | | YCB | | ShapeNet | | YCB | |
| | Min | Mean | Min | Mean | Min | Mean | Min | Mean | Min | Mean | Min | Mean |
| Goal-Auxiliary | 0.015 | 0.045 | 0.020 | 0.047 | 6.621 | 17.989 | 7.610 | 19.465 | 0.023 | 0.065 | 0.028 | 0.069 |
| Goal-Conditioned | 0.016 | 0.039 | 0.025 | 0.045 | 6.987 | 17.638 | 7.821 | 19.864 | 0.025 | 0.056 | 0.033 | 0.071 |

Table 5: Evaluation on grasp prediction performance

Table 5 shows the grasp prediction results. We use the target goals from the expert planner as the ground truth, and compute translation errors in meters, rotation errors in degrees and the point matching distances in meters as used in the loss function in Eq. (2). We compute both the mean error and the minimum error along the trajectories from the expert.

In Table 5, "Goal-Auxiliary" indicates our policy network with auxiliary goal prediction using the aggregated point cloud representation. "Goal-Conditioned" indicates a separate network trained for grasp prediction only, which is used for goal-conditioned models. We can see that the performance of the two networks are quite similar, which demonstrates that our training is able to optimize the auxiliary task well. Overall, jointly training the policy and grasp prediction does not affect the grasp prediction performance. We also observe that the grasp prediction error decreases as the camera gets closer to the object in our closed-loop setting. Note that the regression errors are affected by the discrete nature of the pre-defined grasp set and the multi-modality, especially for certain symmetric objects such as bowls. Figure 7 illustrates some grasp prediction examples on ShapeNet objects.

