# OpenReview forum: "Goal-Auxiliary Actor-Critic for 6D Robotic Grasping with Point Clouds"
_ICLR.cc/2021/Conference — Reject_

### Official Review · AnonReviewer1 · 2020-10-22
**Sound approach with new insights but some design choices don't seem to fit to closed-loop control**

**Rating:** 7
**Confidence:** 4

**Review:**

__Summary__
The paper targets the problem of closed-loop 6D robotic grasping with a parallel gripper based on RGB-D in-hand-camera images. The policy takes an aggregated point cloud (computed from image history) as input (using a PointNet++) and outputs the pose transformation of the gripper. There are several contributions in the specifics of the proposed method. The policy is pretrained using behavioral cloning and DAGGER on known object models, where the expert is composed by a grasp pose sampler and the OMG grasp trajectory planner. Subsequently, the pretrained policy is improved using TD3. Actor and critic networks are each regularized via a loss for solving the auxiliary task of (independently from each other) predicting the final grasping pose. As the goal poses are only available for the expert demonstrations goals are added for the policy roll-out in hindsight, similar to hindsight experience replay, which does not require access to an object model. The approach is evaluated in simulation for grasping YCB and ShapeNet objects with a Franka Emika Panda robot. The evaluation covers ablations of several algorithmical and architectural choices.


__Strong points__
- The paper is well-written
- Thorough and interesting ablations
- Sound approach

__Weak points__
- State-representation might get invalidated when an object is moved, nullifying the biggest advantage of closed-loop grasping
- no comparisons with competing methods (e.g. open-loop 6D grasping + OMG)
- no evaluations in a cluttered environment
- no real robot experiment
- no code


__Recommendation__
I recommend accepting the paper because the system looks sound and promising and is well explained. Some parts, like regularization based on goal pose prediction, are potentially useful also for very different approaches.

__Supporting Arguments__
Overall the approach seems sound and the success rates of around 90% seem good for closed-loop 6D grasping of unknown objects. The ablations that disentangle the contributions of the different design choices are valuable and I was delighted to also find a comparison to the more common and arguably more intuitive approach of using the goal predictions as input to the networks. As the approach and the results are presently sufficiently clear, the paper provides a valuable contribution and should thus be accepted.

Still, there are also several weak points.
- The experiments don't seem to show the benefit of closed-loop grasping by only considering an uncluttered environment and not performing comparisons with open-loop approaches (e.g., [1] or [2]).
- Some design choices do not seem to fit well to closed-loop grasping, which is mainly important for dynamic scenes, e.g. for re-grasping slipping objects. However, in such cases the accumulated point clouds can quickly become invalid, and, furthermore, lower-level control actions might be more appropriate than pose-deltas.
- The approach is not evaluated on a real robot. For all I can tell, the simulation does not account for errors in control, calibration, forward kinematics that are encountered in practice.
- The supplementary did not contain source code. I hope that the authors consider publishing the code after acceptance.

__Questions__
1) How could the proposed method deal with changes in the object position (accumulated point cloud becomes invalid)
2) Did you consider sharing some features between the actor and critic, e.g. parts of the PointNet
3) I assume that the gripper is always closed at the end of a finite-horizon episode, is that correct?
4) I guess the foreground mask is obtained based on the z value of the transformed point cloud. The paper should be more specific here.
5) Please elaborate: "We also empirically observe that the auxiliary task with the bootstrap errors caused by value overestimation."

__Additional Feedback__
If I understand correctly (based on Appendix A) the method uses TD3 instead of "vanilla" DDPG. If this is the case, the main part should also be more concrete.
It's not clear to me what Figure 5 is supposed to show. What does contact-rich mean in this context? Is it about contacts before closing the gripper?

Typos:
extra space in Figure 1: "known objects ."
extra "the" in Section 5.1: "indicates the the aggregated point cloud"

__References__
[1] ten Pas, Andreas, et al. "Grasp pose detection in point clouds." The International Journal of Robotics Research 36.13-14 (2017): 1455-1473.
[2] Mousavian, Arsalan, Clemens Eppner, and Dieter Fox. "6-dof graspnet: Variational grasp generation for object manipulation." Proceedings of the IEEE International Conference on Computer Vision. 2019.

---

> ### Author Response · Authors · 2020-11-16
> **Response to Reviewer 1**
>
> Thank you for your comments and suggestions. We address specific questions below.
>
> Q1: State representation is invalidated when an object is moved.
>
> This is true. For moving target objects, we cannot use the previously aggregated points. We can reset the point cloud from the current time step. Once the object stops moving, we can start aggregating points. According to our experiments, using aggregated point clouds is better than using point clouds from single frames.
>
> Q2: No comparison with open-loop execution with grasp detection methods.
>
> Thanks for pointing this out. We have conducted a comparison with open-loop grasping using the SOTA grasp detection method [1] with OMG. This open-loop method achieves 75.7% success rates in Pybullet, while our method can achieve 94.3% success rate. The open-loop OMG with perfect goals (expert) achieves slightly better success 95.6%. This experiment illustrates grasp detection is a challenging problem. We cannot assume perfect goals in 6D grasping. Closed-loop grasping is critical. Example videos of this comparison can be found in our paper website: https://sites.google.com/view/gaddpg
>
> Q3: No real-world experiments and no cluttered-scene experiments.
>
> Please see our response to all the reviewers. We have conducted the real world experiments, and added some preliminary results for cluttered scenes. Please check the videos in our paper website: https://sites.google.com/view/gaddpg
>
> Q4: No code.
>
> We have uploaded our code to the paper website since October. The official code will be released upon acceptance.
>
> Q5: Low-level control is more appropriate than pose deltas.
>
> We believe that pose deltas have better generalization and simplify the learning compared to using low-level control as actions. Our goal is to learn a policy to grasp unseen objects from different initial states. Learning a mapping directly from sensor input to joint control seems very challenging in this problem. By using pose deltas as actions, we successfully learn an operational space controller using point clouds as input.
>
> Q6: Sharing features between actor and critic.
>
> We experimented with feature-sharing but found it degraded the performance of the critic. Additionally, when using early fusion of input and action for the critic, feature sharing is not applicable.
>
> Q7: Gripper is always closed at the end of the finite-horizon episode. Is that correct?
>
> Yes. The task setup is that the finite-horizon episode terminates when the gripper is close enough to the target. Then a grasp is attempted by closing the finger and lifting.
>
> Q8: How to get foreground masks for object point clouds.
>
> In the simulator, the foreground mask is readily available. For grasping in the real world, we used the unseen object instance segmentation method [2] to segment the target object.
>
> Q9: Elaborate that the goal-auxiliary task helps with value overestimation.
>
> During training, we plotted out the bootstrapped target value from critic and the actual return value. We observed that without grasp prediction, target value estimates grow much more quickly in the beginning of training and eventually lead to large value overestimation of the learner rollouts.
>
> Q10: Isn’t TD3 used instead of “vanilla” DDPG?
>
> Thanks for pointing this out. Yes, TD3 is used in practice instead of DDPG. We consider TD3 as mainly a variant of DDPG so adopt the name DDPG in our method. We will make it clear in the paper.
>
> Q11: What is Figure 5 trying to show? Is it about contacts before closing the gripper?
>
> Correct. We show bad contact with objects before closing in gripper in Fig. 5. Imitation learning alone cannot handle these bad contacts with the object since these are rarely seen in the expert demonstrations. We use RL to help with these scenarios.
>
> [1] Arsalan Mousavian, Clemens Eppner, and Dieter Fox. "6-DOF GraspNet: Variational grasp generation for object manipulation." In IEEE International Conference on Computer Vision (ICCV), 2019.
>
> [2] Yu Xiang, Christopher Xie, Arsalan Mousavian, and Dieter Fox. “Learning RGB-D Feature Embeddings for Unseen Object Instance Segmentation”. In Conference on Robot Learning (CoRL), 2020.

---

> > ### Comment · AnonReviewer1 · 2020-11-17
> > **Thanks for clarification**
> >
> > Thank you for the clarifications and for releasing the code.
> >
> > Regarding Q5) I was trying to make the point that closed-loop grasping is particularly important for reactive grasping, that is, when we want to deal with non-static scenes, slipping objects, etc. However, in such scenarios, the point cloud aggregation and the (lower frequency) position control don't seem suitable.

---

> > > ### Author Response · Authors · 2020-11-17
> > > **Clarification for Q5**
> > >
> > > Thanks for clarifying this question. We agree with Reviewer1 on the imitations of our method in dealing with dynamic scenes. We will consider ways to handle dynamic scenes while keeping the advantages of our state and action representation in future work.

---

### Official Review · AnonReviewer2 · 2020-10-28
**Un-compelling experimental evaluation, novelty concerns**

**Rating:** 6
**Confidence:** 4

**Review:**

This paper tackles the task of closed loop 6-DOF grasping of objects in simulation. The learned policy is a closed-loop policy, in that the gripper pose is continuously adjusted as the gripper approaches the object. The paper employs a combination of imitation learning, reinforcement learning, and auxiliary losses for training this policy. The policy operates upon information from point clouds as observed from a wrist-mounted camera.

Strengths: The paper tackles the important and relevant problem of closed-loop 6DOF grasping. The proposed solution makes sound choices: a) uses fused point-clouds to represent the state, b) use of expert behavior for imitation, c) use of auxiliary losses for training. The paper also does systematic experiments in simulation to judge the importance of the different components.

Shortcomings: While the problem is interesting and important, and the proposed approach is sound, the experiments have entirely been done in simulation. Past work on this topic has studied this problem in the real world. In particular, the paper focuses on the design of closed-loop policies. Closed-loop policies are more relevant when there is noise in the motion of the robot, or there are hard to predict dynamics arising from the interaction of the gripper with the object. These are precisely the aspects that should be studied in the real world, as they are hard to model in order to study in simulation. Thus, it is not clear to me as to what aspects of this paper will be applicable to the study of this problem in the real world.

My second concern is about novelty over past work. All aspects of the technical approach of the paper have been studied in the past. Use of imitation learning and RL together has been studied (eg: DAPG Rajeswaran et al.), use of auxiliary rewards has been studied (eg: UNREAL Jaderberg et al.), use of hindsight experience replay (eg: Andrychowicz et al.). Thus, I am not sure what is the precise technical contribution made in the paper.

Thus in summary, while the paper tackles an interesting and important problem, the problem has only been studied in simulation which makes the application less interesting. At the same time, proposed approach is largely a combination of known techniques in the literature.

Update: I thank the authors for providing clarifications and additional experiments, in particular the comparison to open-loop grasping (SOTA grasp detection method from Mousavian et al.). I still find the technical novelty of the paper limited.

---

> ### Author Response · Authors · 2020-11-16
> **Response to Reviewer 2**
>
> Thank you for your comments and suggestions. We address specific questions below.
>
> Q1: Real world experiments are missing and not sure what parts of the paper is applicable in the real world.
>
> Please see our response to all the reviewers. Our aggregated point cloud representation and the pose delta action are suitable for real world execution with perception noises. Despite the sim-to-real gap in physics and contact modeling, our learned policy shows robustness and generalization over different initial states and shapes in the real world. We wish Reviewer2 can re-evaluate our paper after checking our real-world experiments on the paper website: https://sites.google.com/view/gaddpg
>
> Q2: My second concern is about novelty over past work. All aspects of the technical approach of the paper have been studied in the past.
>
> Our novelty in this paper is on how to solve the 6D grasping problem by combining IL and RL, while most previous works focus on top-down grasping. The state space and the continuous action space are very large in 6D grasping. Vanilla RL cannot learn a useful policy. We propose the GA-DDPG method to tackle this challenging problem. It achieves high success rates in 6D grasping of unseen objects with arbitrary initial state.
>
> We consider our paper to be an application of IL/RL to 6D grasping. We agree with Reviewer2 that combining IL and RL, using auxiliary reward and hindsight experience replay have been studied in the literature, and our method is indeed inspired by these works. However, it is still not trivial to solve the 6D grasping problem with these techniques. The key questions are what input and action representations are useful, how to use an expert planner and how to use goals. Our goal-auxiliary DDPG method is proposed to address these questions for 6D grasping.
>
> We believe that our technical contributions on the aggregated point cloud representation, imitation with an expert motion and grasp planner, goal prediction as an auxiliary task, and hindsight goals for continual fine-tuning would bring significant insights to the community, especially on how to learn closed-loop control policies for 6D grasping.

---

### Official Review · AnonReviewer3 · 2020-10-28
**Weak technical contribution with no major insights from experiments**

**Rating:** 5
**Confidence:** 4

**Review:**

This paper uses several different techniques in IL and RL to improve performance on 6D robot grasping. It uses an expert planner OMG to collect initial data for BC as well as for online IL via Dagger. The uses DDPG to further train as well as fine tune on new unlabeled objects.

The topic is very relevant and of current interest as more real world applications will need more than just 2D grasping that bin-picking has addressed and related work is sufficiently discussed.

The technical contribution seems weak as the paper mostly explores known methods and well-known 'trade-tricks' (goal conditioning or loss on goal) towards a grasping centric problem which is also heavily explored as part of various RL tasks in literature. The main weakness of the work however is the lack of clear motivation for why such a complicated procedure is necessary compared to the expert planner already being used - the experiments aren't designed to address this question.

- Using a planner as an expert for IL is common practice and I don't think counts as a major contribution as presented at the end of the introduction.

- The bulk of IL experiments focus on what input representation is helpful. While it is not surprising that 3D inputs like point clouds would be better for 6D grasping these are more suitable as ablations than main experiments investigating the proposed method itself, compared to other sota approaches learning based or otherwise.

- In several places 'contact-rich' and 'different dynamics' is motivated without clear explanation early on until the experiments identified what the setup was. The former does not seem to be well explored in the experiments, '...especially in those contact-rich scenarios...'. Aren't all grasping problems contact rich (unless only reaching to a pre-grasp is being considered) or were there some new scenarios constructed to specifically study the relative effects of contact?

- Results in table 1 and figure 4 present mean statistic from 3-5 runs. This seems small, variance bands should be shown in figure 4 to see if the small number of sample are sufficient to capture the full picture.

- The problems studied could be addressed by planar grasping as well. Complex setups with clutter, etc would better motivate if the presented approach is able to scale to scenarios where 6D grasping is necessary.

Other comments:

- Does not including the BC loss for some samples in the batch (or between training iteration) cause any discontinuities or make learning unstable, as the loss landscape discontinuously changes?

[Update]
Thank you for the responses and clarifications. I appreciate the additional experiments in the real world and comparisons with the open-loop policy. Novelty still remains a concern however; in using a planner as an IL expert, it isn't clear what was challenging to adopt this strategy for the grasping problem and qualifies as a significant contribution. Additionally, the experiments to study 'contact-rich' and 'different dynamics' problems is unclear; the experiments don't indicate what aspects of the proposed method address these challenges and are able to do so with vision/depth-only feedback (no tactile); also the evaluation in simulation alone is insufficient to study such scenarios. I have updated my score accordingly.

---

> ### Author Response · Authors · 2020-11-16
> **Response to Reviewer 3**
>
> Thank you for your comments and suggestions. We address specific questions below.
>
> Q1: The paper mostly explores known methods and well-known ‘trade-tricks’ towards a grasping problem heavily explored in RL.
>
> Most previous works focus on top-down grasping in RL. We tackle the 6D grasping problem. 6D grasping of arbitrary objects from arbitrary initial states is less studied in RL. The large action and state space in 6D grasping makes the exploration by trials and errors much more challenging.
>
> Our work utilizes common strategies in RL to improve sample efficiency such as using demonstrations and goal-auxiliary tasks. However, we do not think it is trivial to solve the 6D grasping problem by simply combining these well-known tricks. The key questions are what input and action representations are useful, how to use an expert planner and how to use goals. Our GA-DDPG work is proposed to tackle these problems. It achieves high success in 6D grasping of unseen objects.
>
> Q2: Why is such a complicated procedure necessary compared to the expert planner already being used?
>
> The expert planner only works with the full state of the environment. It needs to know the 3D models and poses of the object, while our learned policy works with partial point clouds of objects. This is the main reason we design GA-DDPG to learn a closed-loop grasping policy from partially-observed point clouds.
>
> In our simulation experiments, we showed that learning a policy using the expert planner and behavior cloning only achieves a success rate around 70%. Our GA-DDPG can improve the success rate to 90%. This demonstrates that on-policy learning and goal-auxiliary tasks in our method are necessary. In our supplementary experiments, we also showed that the performance of using the expert planner with a SOTA grasp detection method is worse than ours.
>
> Q3: Using a planner for IL is a common practice.
>
> We agree. But in 6D grasping, the planning problem is not trivial, and humans usually act as experts in previous works. We utilize the recent OMG planner as the expert, which can generate a large number of demonstrations and even label on-policy data for training in simulation where full states are known. Also, the OMG planner selects different goals from different initial states. It enables the learned policy to grasp the same object in different ways. Our way of using the planner has not been explored in previous 6D grasping RL methods.
>
> Q4: It is not surprising that 3D inputs like point clouds would be better for 6D grasping.
>
> This seems intuitive for us too. However, most works in RL use raw RGB or depth images as input to generate grasps and control commands. This is why we think the experiments on input representation are important. We have verified that using the aggregated point clouds as input representations is helpful for 6D grasping. It is worth to investigate different input representations in RL.
>
> Q5: Were there some new scenarios constructed to specifically study the relative effects of contact?
>
> The scenarios of contact in our experiments are on robots touching the object in movement and grasping the object. Contacts and dynamics are critical in 6D grasping. Bad contacts may knock the object down or push the object. In our fine-tuning experiments, we changed the lateral frictions of the objects which affects the success rates of all learned policies. This means that contact models play a role in learning and execution.
>
> Q6: For table 1 and figure 4, is the number of samples sufficient and is variance not shown?
>
> The variance is plotted for figure 4(b). For 4(a), we did not include the variances for aesthetic reasons. In general, the experiments have similar variances for all the policies (around 2% for the final model). We will do more testing runs in the final version.
>
> Q7: The problems studied could be addressed by planar grasping as well and cluttered scenes would be better to study the problem.
>
> We do not agree that planar grasping can deal with these grasping scenarios in our work. For example, there are many cases that it is necessary to grasp the objects from the sides and control the robot hand in 6D due to the initial hand positions. Only for small and light objects on a plane with a fixed hand above, planar grasping is applicable. For cluttered scenes, please see our response to all the reviewers.
>
> Q8: Test if modifying BC updates can cause training instability.
>
> Thank you for pointing this experiment out. In our joint BC+DDPG training, only 30% of the batch has the BC loss. We observed that the specific batch portion and loss scale balance do not affect the performance dramatically. We have also experimented with completely removing BC loss, and it causes some training instability possibly due to the conflict between the gradient of the critic and the BC gradient. Lastly, note that TD3 only applies policy gradience every two updates. BC indeed plays a key role in our work for learning efficiency and generalization.

---

### Author Response · Authors · 2020-11-16
**Response to All Reviewers**

We thank all the reviewers for their valuable comments and suggestions. We will revise our paper accordingly, and will release our code for reproducibility. We address common questions from the reviewers below.

Q1: Real world experiments

Due to the pandemic, we could not finish the real-world experiments when submitting the paper. We have conducted the real-world experiment and added it to the paper website: https://sites.google.com/view/gaddpg

We directly applied our policy learned in PyBullet to the real world. Surprisingly, it works pretty well on 6D grasping of unseen objects. We believe this is due to our point cloud representation which has a smaller domain gap compared to using images. To obtain the segmented point cloud of an object, we applied an unseen object instance segmentation method. The policy outputs the delta 6D pose of the robot gripper at each time step, which is used to control the robot for grasping.

For the 9 YCB objects in our experiments, the robot grasped each object 5 times. The learned policy achieves above 70% success rate. Most failures in the real world are due to slippery and mismatch in physics between simulation and the real world, where the same hand pose can succeed in simulation but fail in the real world. We plan to fine-tune the policy in the real world as a future work. The robot also tried grasping 10 unseen objects. It only failed one among the 10 objects. Please see the videos on the paper website. We will add details about the real world experiments to the paper.

Q2: Handle cluttered scenes

We are working on extending the closed-loop grasping to cluttered scenes. Another problem to solve in cluttered scene grasping is collision avoidance with other objects. We consider GA-DDPG to be a policy that can be applied when the robot gripper is already close to the target object. We are working on learning a policy to avoid other objects and grasp the target object jointly for cluttered scenes. Some initial results are shown in a new video on the paper website: https://sites.google.com/view/gaddpg

---

### Author Response · Authors · 2020-11-19
**Paper Revision**

We thank all the reviewers for their insightful comments again. We have revised the paper as suggested by the reviewers. The main paper has 9 pages now. We summarize the major changes as follows:
- We added details about the real-world experiments
- We added a comparison with an open-loop 6D grasping baseline in the experiments
- We added a discussion on limitations of our method in the Appendix
- We fixed typos in the paper

---

### Decision · Program_Chairs · 2021-01-07
**Final Decision**

**Decision:**

Reject

**Comment:**

The paper got mixed reviews ranging from 5 to 7. The main concerns of the reviewers were the missing novelty as the paper combines different well known methods for a given problem, so there is no big algorithmic contribution. The presented pipeline for closed-loop grasping using imitation learning from a planner, Dagger and subsequent deep RL with TD3 is a straightforward, but sound and intuitive combination of algorithms to address the problem of closed loop grasping.  The presented results and ablation studies also motivate these algorithmic choices. In the rebuttal the authors addressed most concerns regarding the experiments (missing comparisons to open-loop grasping and real world experiments), but more real world experiments would be necessary to evaluate the effectiveness of the approach.

This is a borderline paper were  I unfortunately have to recommend rejection due to the missing algorithmic contribution, a major requirement for ICLR. The paper would  however fit very well to a robotics conference and the authors are  encouraged to resubmit the paper the venues such as RSS or CoRL.